# Seaweed Polysaccharide Based Products and Materials: An Assessment on Their Production from a Sustainability Point of View

**DOI:** 10.3390/molecules26092608

**Published:** 2021-04-29

**Authors:** Nishith A. Chudasama, Rosy Alphons Sequeira, Kinjal Moradiya, Kamalesh Prasad

**Affiliations:** 1P. D. Patel Institute of Applied Sciences, CHARUSAT Campus, Charotar University of Sciences and Technology, Changa 388421, India; chudasamanishith@gmail.com; 2Natural Products & Green Chemistry Division, CSIR-Central Salt and Marine Chemicals Research Institute, Gijubhai Badheka Marg, Bhavnagar 364002, India; fleviarosy@gmail.com (R.A.S.); kinjalmoradiya7@gmail.com (K.M.); 3Academy of Scientific and Innovative Research (AcSIR), Ghaziabad 201002, India

**Keywords:** agar, carrageenan, alginic acid, extraction, chemical modification, application, ionic solvent

## Abstract

Among the various natural polymers, polysaccharides are one of the oldest biopolymers present on the Earth. They play a very crucial role in the survival of both animals and plants. Due to the presence of hydroxyl functional groups in most of the polysaccharides, it is easy to prepare their chemical derivatives. Several polysaccharide derivatives are widely used in a number of industrial applications. The polysaccharides such as cellulose, starch, chitosan, etc., have several applications but due to some distinguished characteristic properties, seaweed polysaccharides are preferred in a number of applications. This review covers published literature on the seaweed polysaccharides, their origin, and extraction from seaweeds, application, and chemical modification. Derivatization of the polysaccharides to impart new functionalities by chemical modification such as esterification, amidation, amination, C-N bond formation, sulphation, acetylation, phosphorylation, and graft copolymerization is discussed. The suitability of extraction of seaweed polysaccharides such as agar, carrageenan, and alginate using ionic solvent systems from a sustainability point of view and future prospects for efficient extraction and functionalization of seaweed polysaccharides is also included in this review article.

## 1. Introduction

Polysaccharides are high molecular weight macromolecules present in all living systems including plants and seaweeds. They act as structural materials and as suppliers of water and energy [1]. Due to the structural and morphological versatility, they find applications in various industries ranging from food to pharmaceuticals. Most of the polysaccharides are water-soluble (at elevated temperatures) or they can be swollen in water under ambient conditions giving the formation of colloidal, highly viscous solutions or dispersions with pseudoplastic flow properties. The inherent functional properties of the polysaccharides such as thickening, water holding and binding, stabilization of suspensions, emulsions, gelling, etc., make them useful in various applications [2]. In several occasions, tailor made functionalities for the polysaccharides can be achieved as well. Among the polysaccharides, seaweed-based polysaccharides such as agar, agarose, alginic acid, carrageenan, ulvans, and fucoidans are exploited for various commercial applications. The polysaccharides with specific and distinguished characteristics are isolated from the vast marine source and such polysaccharides are absent in terrestrial plants. These algal polysaccharides serve as fascinating tools for therapeutic and industrial applications, which include nutraceuticals, pharmaceuticals, cosmeceuticals, and functional foods. At present such bioactive sulphated polysaccharides are used as formulations in feed, food, and pharmaceutical applications but their various characteristic biological capacities attract researchers to explore their applications in a number of other fields [3,4]. Marine polysaccharides have also been explored as efficient drug delivery systems combining the advantages of micelles and natural biopolymers showing excellent compatibility, biodegradability, non-toxicity, non-immunogenicity, extended blood circulation time, and better drug loading capacity [5,6,7]. Polysaccharides from the marine sources have found their value addition due to their unprecedented therapeutic properties and hence are exploited in tissue engineering, stent coating, and biomolecules immobilization [8].

As mentioned above, several seaweed polysaccharides are used in medicinal and pharmaceutical fields to impart various biological activities. Seaweed polysaccharide based nanoparticles, microspheres, and gels were found to have sustained and controllable drug delivery potential for anticancer and anti-inflammatory drugs [9]. Several sulphated polysaccharides were found have cytotoxic properties and were used as antiviral substances against respiratory syncytial virus (RSV), herpes simplex virus (HSV) types 1 and 2, and human immunodeficiency virus (HIV). Carrageenan has been demonstrated as a potential agent in vitro to impart antiviral activity like blocking of HIV and other sexually transmitted diseases (Carraguard (vaginal microbicide)) [10]. Biomedical and biological applications of seaweed polysaccharides, i.e., alginate, carrageenan, fucoidan, and ulvan displays in drug delivery, tissue engineering, biosensor, and wound healing because of their gel forming properties and capacity of inducing important differentiation in stem cells [11]. Alginate is used for both immediate (for rapid absorption) and sustained (for reproducible and kinetically predictable) release of drug. Alginic acid and sodium salt of alginic acid are used as tablet disintegrant and tablet binding agent respectively for immediate drug release. The chitosan-alginate composite and calcium salt of alginic acid were used in wound healing and tolerance for diabetic foot lesion. In such applications, calcium from the alginate salt and sodium of the wound’s exudates goes through ion exchange and form sodium alginate soluble gel, while free calcium ions help in clotting and endowing the dressing [12]. Alginate and alginate beads were used in various treatments like in diabetes treatment by the transplantation of chondrocytes, hepatocytes, and islets of langerhans, treatment of gastroesophageal reflux disease and heartburn [13,14,15,16,17]. Combination of alginate, chitin/chitosan, and fucoidan gave a hydrogel sheet having favorable properties for wound healing in rats [18]. Similar to alginates, carrageenan is also used in drug delivery applications in various forms such as beads, gelling agent, nanoparticles, nano stabilizer, micro stabilizer, microspheres, and in microcapsules [19,20]. In the treatment of hypercholesterolemia, carrageenan can be used for sustained fluvastatin drug release and formulation [21]. The sustained release of ovalbumin macromolecule by the nanoparticles of chitosan-carrageenan is also reported [22].

As evident from the above studies, seaweed polysaccharides are explored for their potential application as pharmaceutical drugs to treat number of ailments and hence it becomes very important to study their pharmacokinetics. This focuses study on the drug behavior after its administration in the body systems that include absorption, distribution, metabolism, and excretion (ADME). This helps to understand pharmacological activity at the molecular level to determine correct doses, treatment methods, and specific drug applications [23]. The parameters, which are important to gain knowledge on changes in drug concentrations with ADME, are apparent half-life of elimination (T1/2), the area under the curve (AUC), clearance (Cl), maximum concentration (Cmax) and time at which Cmax is observed (Tmax), median residual time (MRT), the high volume of distribution in the blood (Vss), and bioavailability (F) [24]. Several analytical methods like biomarker assay, anti-Xa activity, gas chromatography, ELISA, HPLC, etc., are used for the pharmacokinetic study of fucoidan. Furthermore, the pharmacokinetics of alginates and fucoidan seaweed polysaccharides are explored so far [23]. Despite the extensive exploitation of seaweed polysaccharides for their pharmaceutical applications, the pharmacokinetics of the same has not been explored much. Thus, an increase in this field of research is expected in the near future.

Considering the potential of the large scale application of seaweed polysaccharides in various fields, it is important to understand the market positioning of the products from sustainable exploitation point of view. *Kappaphycus alvarezii* and *Eucheuma denticulatum* are the two seaweed species that contribute to 88% of raw material for carrageenan production. Malaysia, Indonesia, and Philippines produce around 1,20,000 tons of the polysaccharide per year [25]. The global carrageenan market was estimated to be around 931.6 million USD in 2020 and its valuation is expected to reach 1.2 billion USD at a CAGR (compound annual growth rate) of 5.6% by 2025 [26]. China is the world leader in agar agar production with a harvest of 2.7 million tonnes of farmed *Gracilaria* seaweed (main source of agar) in 2015 [27]. In 2015, the global agar market was estimated to be USD 214.98 millions, which is further anticipated to grow at 4.9% CAGR from 2016 to 2025 [28]. The alginate production is estimated to be around 30,000 metric tonnes annually, which comes from farmed brown seaweed of the genera *Laminaria* and *Macrocystis* [29]. The global alginate market size was valued at USD 728.4 million in 2020 and is expected to grow at a compound annual growth rate (CAGR) of 5.0% from 2021 to 2028 [30]. The source of certain seaweed polysaccharides and their chemical structure is depicted in Table 1 and Figure 1 respectively.

### 1.1. Agar and Agarose

Red seaweeds (*Rhodophycea*) such as *Acanthopeltis* spp., *Campylaephora* spp., *Ceramium* spp., *Gelidium* spp., *Gracilaria* spp., *Pterocladia* spp., etc., are the major source of agar or agarose. Apart from the use of agar in the food and beverage industries, this phycocolloid is also used for bioengineering and biomedical applications as a gelling agent. Chemically 1,3-β-*D*-galactose and 1,4-α-*L*-3,6-anhydrogalactose are the basic repeating unit of agar or agarose [39] (Figure 1).

Agar is isolated from red seaweed (agarophytes) employing various methods such as freezing–thawing, extrusion using a hydraulic press, and solvent precipitation as depicted in Scheme 1. Due to the biocompatible and non-toxic properties, it is widely used as a thickener, stabilizer, and emulsifier in the food and beverage industries. Agar based films with enhanced shelf-life are explored for their suitability to replace plastic based packaging materials [40]. It is essential to understand the enzymatic mechanisms and agar biosynthesis to help in selecting seaweed materials with extended gelling properties utilizing molecular markers [41]. The rheological and thermal properties of agar change upon alkali treatment and agaropectin enhances the gelling ability of agar [42]. Agar hydrogels were formed by mixing of agar-agar with different compounds such as glycerin, sorbitol, sodium citrate, and sodium chloride with varying concentrations and their rheological behavior was studied. Such studies are important to design hydrogels with required characteristics for specific applications [43]. Agar hydrocolloid is used as thickening and gelling agents for food applications and soft capsule preparation methods are also developed using agar as its base [44,45]. Agarose microparticles are explored to develop textural functionalities in beverages from liquid to fluid gels [46]. Agarose is used in the form of gel-based separation phase for microextraction due to easy fabrication, high inertness, and biodegradability [47].

Furthermore, derivatives of agar are used in various applications such as sweetening agent, in bacterial culture, pH-responsive/stability materials, fluorogenic material and for controlled release applications, and in the preparation of self-assembled nanomaterials [48,49,50,51,52,53,54,55,56,57].

Agarose is purified agar that is commonly used in the gel electrophoresis process for the separation and purification of nucleic acid and proteins. To prepare it, agar needs multi step further purification to remove the charges to make agarose neutral. Considering the high cost involved in making agarose from agar, a cost-effective extraction method using surfactant was developed [58]. In this method, molecular biology grade agarose can be selectively isolated from the agarophyte extract as shown in Scheme 1.

### 1.2. Alginic Acid

Alginic acid or alginates are commercially important polysaccharides having several applications in the food, beverage, and pharmaceutical industries. Alginic acid in the form of sodium, ammonium, and polypropylene glycol alginate (PGA) is used in various applications. Alginates are the major structural components (ca. 40% of total dry mass) of the cell walls of brown seaweed (*Phaeophyceae*) and they play a crucial role in maintaining the structure of the algal tissue. α-*L*-Glucopyranosyl and β-*D*-mannopyranosyl are the two monomeric units that are arranged in three distinct patterns to form the backbone of linear block copolymer alginic acid by guluronic acid (G), mannuronic acid (M), and intermediate composition (MG) as shown in Figure 1 [59].

The brown seaweeds are one of the most abundant seaweeds found in many parts of the world. The giant kelp *Macrocystis pyrifera* is harvested mechanically using special ships on the west coast of North America. These seaweeds have very high growth rate and hence they can be harvested several times a year. Other species of the brown seaweeds, which are relatively smaller in size in comparison to *Macrocystis* spp., are harvested semi-mechanically using fishing boats or manually. Several brown seaweeds such as *Ascophyllum*, *Laminaria*, *Macrocystis*, and *Sargassum* species are used for the extraction of sodium alginate employing calcium alginate and/or alginic acid processes as shown in Scheme 2.

The key properties of alginates that make them useful are their ability to increase the viscosity of fluids and the ability to form gels and films. They exhibit excellent properties such as water solubility, biodegradability, film forming ability, and biocompatibility. These unique properties of the natural polysaccharide make it useful in fields of healthcare food industry, catalysis, for water treatment, packaging, and immobilization of cells [60,61]. This biodegradable polymer has also found its application in tissue engineering and preparation of biomaterial scaffolds that are very important for rendering medical needs [62]. Recently, alginic acid has been studied for its structural, molecular, and functional abilities such as tablet ability, elasticity, deformation ability, disintegration ability, and compressibility [63].

Additionally, alginates are used in textile printing, as release agents for paper, welding rods, binders for fish feed, etc., and in medical and pharmaceutical applications [64].

### 1.3. Carrageenan

Carrageenans are another class of seaweed-based polysaccharides that have several applications in various industries. There are several types of carrageenans available with different chemical structures and properties. The primary source of carrageenan is red seaweeds such as *Sarcothalia crispate*, *Gigartina skottsbergii*, *Eucheuma denticulatum*, *Kappaphycus alvarezii,* and *Chondrus crispus*. The carrageenan composition differs in different species of red seaweeds (carrageenophytes). ι-Carrageenans,κ-carrageenans and λ-carrageenans are the major forms of carrageenans. *D*-Galactose-4-sulphate and 3,6-anhydro-*D*-galactose-2-sulphate, *D*-galactose-4-sulphate and 3,6-anhydro-*D*-galactose, and *D*-galactose-2-sulphate and *D*-galactose-2,6-disulphate (Figure 1) are the basic disaccharide repeating units of ι-carrageenans, κ-carrageenans, and λ-carrageenans respectively. In the presence of metal ion, ι-carrageenans and κ-carrageenans give the formation of a gel but λ-carrageenan does not form a gel [39]. Carrageenan is isolated from carrageenophytes following different methods such as the freeze–thawing method, KCl precipitation method, solvent precipitation method, etc. (Scheme 3).

Carrageenans are used to impart thickening, stabilizing, and gelling properties in various food and beverage formulations. These phycocolloids also find application for immobilization of biocatalysts, in toothpaste as a stabilizer, in air freshener gels, pet food, and meat products [64].

In recent times, carrageenan-based biomaterials are gaining attention due to their multifunctional properties such as biodegradability, biocompatibility, non-toxicity, antiviral, antibacterial, anticoagulant, antioxidant, antitumor, and immunomodulating properties [65,66]. Carrageenan has been widely exploited as food additives and several carrageenan based biomaterials for drug delivery applications have also been developed. These materials are also studied for their adverse effect on biological systems. The pH sensitivity and adhesive properties play an important role in preparation of such biomaterials [67,68,69]. These polysaccharides are a source of sustainable and renewable polymers that can be employed for film formation and as coating materials. Blending these with other materials to form composites enhance their film properties for potential applications. Such composites exhibit enhanced tensile strength and reduced hydrophilicity [70]. These sulphated polysaccharides have been exploited for their use in the preparation of oral release tablets as a novel extrusion aid for the production of pellets and as a carrier stabilizer in micro/nanoparticles system. Due to its therapeutic properties, it is also used in tissue regeneration and cell delivery [71]. They are used as control release vehicles, gelling agents, encapsulating agents, beads, films, and can efficiently encapsulate flavors, fragrances, enzymes, and probiotics [72]. A recent study shows that these polysaccharides have anti-cancer activity thereby improving immunity and exhibit chemotherapeutic effects [73].

## 2. Chemical and Physical Modification of Seaweed Polysaccharides

Over the past several decades, biopolymers have received special attention for their application in the medical, biomedical, and chemical industries. Among the natural polymers, chitin structures are popularly used in medical applications due to their absorbable nature by human tissues. Due to the biocompatible nature and compatibility among natural polymers, chitosan-collagen composite films were prepared for use as an artificial replacement of human skin. The biodegradability and edibility of such polymers make them more useful in such applications [39,74]. However due to the increasing scarcity of bioresources for material generation for the future, seaweeds are being given special attention to use them as suitable biomass resources for various products and materials. The seaweeds are associated with several advantages such as excellent growth rate, cultivable, no need for land and fertilizers for cultivation, etc. Among the seaweed-based polysaccharides agar/agarose, carrageenan, fucoidan, ascopllyan, prophyran, and alginates are the most exploited biopolymers for products and materials. The polysaccharides are chemically and physically modified to generate new functional derivatives to make them suitable for targeted applications. Different functional groups (e.g., amine, carboxylic acid, amide, thiol, etc.) were used to modify polysaccharides for various new applications.

Agarose was modified by substituting a new functional group in the primary alcohol group at their C-6 carbon of (1,3) linked β-*D*-galactose moiety. The derivatives thus obtained were characterized thoroughly using competent analytical tools. Usually, organic and aqueous solvents were used for the modification. In organic solvents, DMF and DMSO were used for the reactions. Most of the reactions were done under microwave irradiation for improved yield and to shorten the time durations. The modification reactions involved esterification [49,52], amination [56], amidation reactions [56], -C-N- bond formation [53,54,55,57], etc. (Figure 2). These modified agarose derivatives have new functional properties such as enhanced fluorescence emission, sweetness properties, cationic properties for gene/drug delivery, ability to form fluorescence hydrogel, highly stable hydrogel formation, stable in different pH solution, self-assembled nanoparticles, etc.

Apart from the methods mentioned above for functionalization, grafting, and crosslinking can be used to chemically modify polymers. Grafting a polymerizable synthetic moiety on a natural polysaccharide followed by polymerization is a way of creating large molecules, which have some of the properties of each of the polymers. The primary focus of carrying out these types of grafting reactions is to obtain products having good water absorption properties, new polymers that can form sheets [75]. Due to the bulky industrial application of starch and cellulose in textile, paper, and food industry, they are being grafted or crosslinked to produce derivatives having different functional properties [76].There are several reports of chemical modification of agar and agarose by inducing hydrophobic groups such as alkyl, acetyl, and hydroxy alkyl groups to decrease gelling and melting temperature, which is very crucial for bacteriological applications of the polysaccharide [77]. Low gel strength agar gel is prepared by reacting high gel strength agar with salts of weak acids viz., citric and ascorbic acids. Such soft gels are mainly used as massage gels, skin moisturizers, or as an active carrier for pharmaceuticals, which have to be applied through skins [78]. κ-Carrageenan based super absorbent hydrogel was synthesized through graft polymerization of acrylic acid on the biopolymer back bone in presence of crosslinking agent (N, N’-methylene bis acrylamide) and an initiator (ammonium persulphate) [79]. Due to the compatibility of carrageenan with inorganic materials, sol–gel biomaterial based on the polysaccharide and silica was prepared [80]. Physical modification of agar gels by interacting agar with ionic and nonionic surfactants to study the gel inhibition effect is studied [81].

Numerous chemical modifications have been proposed to modify the physicochemical properties of carrageenan. Most of the modifications aim to induce new properties by substituting the primary alcohol group present in the *D*-galactose-4-sulphate moiety of the biopolymer with a new functional group. The modification reactions involve free radical graft-copolymerization, sulphation, acetylation, and phosphorylation [82,83,84] (Figure 3). All of the carrageenan derivatives have new functional properties such as improved absorptivity, gel strength, mechanical, magnetic, rheological, water-holding capacity, swelling properties, antioxidant activity, biodegradability, stability in different pH region, storage, thermal stability of the enzymes upon encapsulation, etc., which broaden the application area of the polysaccharides [85].

Alginates are important polysaccharides that form hydrogels at mild temperature and pH making them valuable industrial pharmaceutical excipient [86,87,88]. Biomaterials like nanospheres are prepared using alginates that exhibit drug delivery capacity and facilitate cell imaging. Alginate derivatives are also used for preparation of plastic materials with tunable organosolubility and thermal properties [89,90,91]. Catalytic hydrogenation of the polysaccharide leads to sugar alcohols that may provide opportunities to diversify biomass resources [92]. Alginic acid can be chemically modified by attaching various moieties at the end of the carboxylic acid group of guluronic acid (G) or mannuronic acid (M). The obtained derivatives were characterized thoroughly using various analytical tools. Usually, the sodium salt of alginic acid was used for the modification because of the good solubility of sodium alginate in water. Most of the reactions were done under microwave irradiation for improved yield and to shorten the time duration. The modification reactions involved esterification and amidation (Figure 4). These modified alginic acids have new properties and has the potential to be used in fluorogenic metal scavenging [93,94], thixotropic material [95,96], efficient photosensitizing material [97], etc.

In another attempt, chemical reaction of sodium alginate with o-aminobenzoic acid and m-aminobenzoic acid in the presence of 1-ethyl-3-[3-(dimethylamino) propyl]-carbodiimide hydrochloride (EDC) gave formation of amide derivatives of the biopolymer that could form viscous hydrogel systems with thixotropic behavior [96]. The soft gel thus formed turned into a flowing liquid on gentle stirring making it suitable for possible application as a delivery system or sprayable gel material for transporting small active molecules to the targeted locations in health and personal care formulations (Figure 5).

Considering the suitability of grafting copolymerization reactions for the preparation of new derivatives of seaweed polysaccharides, alginates were also grafted with suitable polymerizable monomers such as acrylonitrile, methacrylate, acrylamide, etc., to prepare new materials or products of the polysaccharide. In one of the endeavors, polyacrylonitrile grafted agar/sodium alginate (Agar/Na-Alg-graft-PAN) was synthesized in an aqueous medium under reflux conditions in the presence of potassium persulphate as a free radical initiator [98]. The resulting polymer was found to have improved swelling properties in water and stability in acidic pH (Figure 6).

## 3. Ionic Solvents

### 3.1. Ionic Liquids and Deep Eutectic Solvents

Ionic liquids (ILs) are molten salts consisting of ions with a melting point lower than room temperature or below the boiling point. ILs have properties such as non-flammability, low vapor pressure, low melting point, excellent thermal and electrochemical stability, high conductivity, high stabilization of specific solutes and the ability to recycle [99]. By using various combinations of cations and anions, ILs have a wide range of applications in the fields of lubricants and additives (lubricants and fuel additive) [100,101], electro-elastic material (artificial muscles and robotics) [102], analytics [103,104,105], solvents for processing [106,107,108,109,110,111,112,113,114], liquid crystals (displays) [115,116], heat storage (thermal fluids) [117], electrolytes [118,119,120], and separation (gas separations, extractive distillation, extraction, and membranes) [121,122,123].

Deep eutectic solvents (DESs) are formed from hydrogen bond acceptor (HBA) and hydrogen bond donor (HBD) by simply mixing them followed by heating [124]. DESs can be used as an alternative of ILs because DESs have similar physicochemical properties as ILs (low vapor pressure and melting point, non-flammability, high thermal and chemical stability, high dissolution ability, and the ability to recycle) [125,126]. However, DESs are a bit different from ILs because DESs are not completely composed of ionic species and it can also be produced from non-ionic components [127]. The melting point of an individual hydrogen bond donor and hydrogen bond acceptor of deep eutectic solvent is considerably higher because of the charge delocalization due to hydrogen bond between HBA and HBD. The dissolution of biopolymers such as DNA also follow a different mechanism in both the solvent systems (ILs and DESs) [128].

### 3.2. Sustainable Extraction of Seaweed Polysaccharides Using Ionic Solvents

Although polysaccharides are present in plants or seaweeds in reasonably in good proportions (15–60% *w*/*w*) but their extraction from the bio resources is very crucial and deciding factor for the commercial viability for the polysaccharide-based industries. As described above, the extraction of polysaccharides from seaweed by conventional extraction process involve number of steps also produce effluents. Excessive amounts of alkali and acids are commonly used to break the cell walls of the seaweeds and the unutilized chemicals remained as effluent at the end of the process. Moreover, conventionally one of the commercially most important seaweed polysaccharide known as agar is extracted from agarophytes employing alkaline pretreatment of the seaweed followed by autoclaving and subjecting the aqueous extract to several cycles of freeze–thaw to isolate the product followed by purifying the product further through solvent/chemical treatment and/or chromatography to eliminate residual impurities (Scheme 1) [129]. The agar thus obtained is further purified employing chromatographic techniques to obtain agarose. Considering the lengthy purification and expensive isolation processes and requirement of expensive infrastructure, the process for the isolation of agar and preparation of agarose is not attractive for small-scale industries. Hence, reduction of operational steps and lower usage of chemicals or use of recyclable processing solvents may a sustainable approach for the isolation of seaweed polysaccharides. Due to the very good dissolution, extraction, and reuse efficiency of ionic liquids (ILs), few of the imidazolium and ammonium based ILs were found to be suitable for the extraction of agar from *Gracilaria* dura, a red seaweed [130]. More specifically the agarophyte was treated with 1-ethyl-3-methylimidazolium acetate under microwave conditions and agarose was isolated by precipitation in methanol. However, in order to make a process more sustainable the use of chemicals in the process and number of operational steps must be reduced. Selective coagulation of the targeted molecule using a single step treatment is always a better way to isolate products from the complex natural matrices. In one of the attempts, a biobased ionic liquid namely choline laurate was found to have a selective affinity towards agarose present in the hot seaweed extracts resulting in eventual selective precipitation of the biopolymer as depicted photographically in Figure 7 [131]. This method may be considered as sustainable since agarose was directly obtained from the seaweed extract without involving any purification stage of agar and freezing–thawing step that are energy intensive (Scheme 1). The ionic liquid was recyclable and reusable for subsequent batches of experiments. Isopropyl alcohol (IPA) used to wash the final product was also recovered by distillation and reused in the process. The process ensures lower energy inputs in the form of presence of lesser number of operations resulting lesser duration of the process. Further, recyclability and reusability of solvents ensures use of lesser amounts of chemicals and this make the process sustainable.

Similar concept of preferential coagulation of κ-carrageenan from the water extract of *Kappaphycus alvarezii* was demonstrated as shown in Figure 8 [132]. Herein, unlike the conventional process of extraction (Scheme 2), no alkali treatment of the seaweed was done and no acids were used during extraction. The polysaccharide was preferentially coagulated using biobased ILs such as choline capriate and choline laurate. IPA used to wash the product was also half to the amount used conventionally in such isolations. Due to the bio origin it is much safer to dispose the ionic liquid wastes produced in the process. These merits make the process sustainable.

Furthermore, in order to replace energy intensive autoclaving of seaweeds to facilitate extraction under pressure and considering several advantages associated with deep eutectic solvents (DESs), few hydrated DESs were used to extract κ-carrageenan from *Kappaphycus alvarezii* [133]. As shown in Figure 9, *Kappaphycus alvarezii* powder was soaked in several hydrated DESs followed by 1 h extraction at elevated temperature (85 °C) followed by centrifugation and washing with IPA resulting isolation of κ-carrageenan more efficiently in comparison to the conventional extraction methods [133]. The yield of the polysaccharide was about 60% in comparison to about 36% by conventional process. Unlike conventional process of isolation of carrageenan, herein no alkaline treatment or autoclaving was done and use of hydrated DESs ensure the usage of lower amount of chemicals in the process. The effective DESs used are of bio-origin and hence the disposal is not going to create any environmental issues. These merits make the process sustainable in nature.

### 3.3. Modification of Seaweed Polysaccharides in Ionic Solvent

Ionic liquids are considered as a good choice for chemical transformations. ILs have a good ability to solubilize polysaccharides such cellulose, starch, and many more resulting formation of homogeneous conditions suitable for chemical derivatizations [134]. In a large variety of polysaccharide modification reactions, imidazolium-based ionic liquids are very much appropriate because they are chemically inert for a number of organic reactions [135]. In homogeneous reaction conditions of polysaccharides such as acetylation, carboxymethylation, esterification, etherification, regioselective enzymatic acylation, tosylation, etc., ionic liquids are emerging as ideal solvent systems. Further, the characteristics of the solvent such as low vapor pressure, recyclability, high boiling point, etc., are added advantages for carrying out such modifications. Acetylation and tosylation of cellulose in 1-allyl-3-methylimidazolium chloride indicate suitability of the ILs for chemical modification of natural polymers [136,137]. The carboxymethylation and esterification of cellulose were also positively accomplished in 1-N-butyl-3-methylimidazolium chloride ([C4mim]+Cl-) [138]. Enzymatic regioselective acylation of unprotected monosaccharides in ionic liquids is also reported [139].

## 4. Functionalization of Seaweed Polysaccharides by Interaction with Other Bio Macromolecules

Numerous research have studied the interaction of biopolymers for at least 50 years. The predominant interactions studied are protein–protein interaction, protein–polysaccharide interaction, DNA–protein interaction, DNA–polysaccharide interaction, DNA–DNA interaction, DNA–carbohydrate interaction, etc. [140]. Intermolecular interactions between biopolymers produce enhanced functional properties in comparison to the individual biopolymers. These interactions depend on the different intermolecular forces like covalent interactions, Van der Waals interactions, hydrogen bonding, ion-bridging, electrostatic forces, and hydrophobic interactions between two biopolymers [141], and some other factors like mixing ratio, molecular concentration, pH, ionic strength, charge density, molecular conformation, charge distribution, molecular weight, shear, pressure, temperature, and acidification [142]. These complexes, obtained from the interaction of biopolymers are used in dairy products, to control the shelf-life, texture, and structure of semisolid foods through their gelling/thickening behavior and surface properties, separations, drug carrier, chiral sensing applications, etc. [143,144,145].

Considering the application of seaweed polysaccharides in food and beverage industries, the interaction of agarose derivatives (6-aminoagarose succinate half-ester derivative) with protein and DNA in various pH regimes was studied. Modified agarose showed different complexation and decomplexation with bovine serum albumin (BSA) protein and DNA in pH 5.2, pH 6.8, and pH 10.0 [146]. Further modification of the agarose derivative with PEG resulted in formation of PEGylated amphoteric agarose, which showed interactions with BSA [147]. Furthermore, agarose was derivatized to generate two fluorescent aromatic agarose amino acid nanoconjugates, which also showed interactions with BSA [148].

## 5. Future Prospect

As discussed above due to several problems associated with the conventional process of extraction of seaweed polysaccharides it is a pressing need to develop a sustainable process for the extraction of the polysaccharides with the involvement of lesser energy inputs in the form of lesser number of steps and lower time durations. The processes also must have usage of a lower amount of chemicals and must produce a lower amount of effluents to meet sustainability. As discussed in this article the ionic liquids and deep eutectic solvents may be considered as sustainable solvent systems for the extraction of seaweed polysaccharides. It is demonstrated that few of the polysaccharides can be extracted with higher yield, lower effluent production using ILs and DESs without using acids and alkali. The high boiling point and recyclability of the solvents also ensure lower usage of chemicals and lower production of volatile organic compounds (VOCs), which make the processes more sustainable. However, such work needs to be scaled up and techno-commercial feasibility of the processes should be evaluated for commercial exploitation in a sustainable manner. Other commercially important seaweeds such as ulvans and fucoidans are not being extracted using any ionic liquids or deep eutectic solvents. Considering the tremendous dissolution ability of the ILs/DESs, they should be used for the extraction of these polysaccharides as well. There is a need for the development of commercially viable extraction processes based on ionic liquids and deep eutectic solvents for the commercially important seaweed polysaccharides. Furthermore, as discussed above, seaweed polysaccharides and their derivatives have a very large range of applications. To target new applications or to improve the functionalities in existing applications, it is important to synthesize new derivatives of the polysaccharides. There are many polysaccharide derivatives still left to be synthesized and such derivatizations should be attempted to further add value to the seaweed derived products and target application in the industries. Moreover, there is a need for the development of commercially viable synthesis processes and the use of recyclable solvents like green neoteric solvents may be encouraged in industrial applications for such syntheses. The interaction of polysaccharides with other protein and DNA may led to the formation of new derivatives having improved pH-responsive cationic/anionic drug capturing and releasing properties, protein binding, chiral sensing, and separations and hence derivatization of polysaccharides by interaction with other macromolecules may be targeted in order to develop cost effective sustainable derivatization methods and products.

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
