# Peer review of "Seaweed Polysaccharide Based Products and Materials: An Assessment on Their Production from a Sustainability Point of View"

_molecules, 2021, doi:10.3390/molecules26092608_

Round 1

Reviewer 1 Report

In the manuscript “Seaweed polysaccharide based products and materials: an assessment on their production from sustainability point of view” by Chudasama et al. the authors review the literature articles concerning the production and usage of different polysaccharides obtained from sea products.

The paper is well written, clear but some corrections are needed:

  1. It will be of interest if the authors can indicate the total amount of produced marine polysaccharides in the last years (tones/year)
  2. the number of recent articles is very scarce therefore some recent references must be added, such as: https://dx.doi.org/10.3390%2Fmolecules25143152; https://doi.org/10.3390/polym13030477; https://doi.org/10.1016/j.progpolymsci.2017.06.001; https://doi.org/10.1007/s12272-017-0958-2; https://doi.org/10.3389/fpls.2020.554585; https://dx.doi.org/10.3390%2Fmd18010007; http://dx.doi.org/10.3390/md18120658; but there are many more and the authors must indicate their originality in comparison with ones of these reviews.

Author Response

Response to the Reviewer’s comments: molecules-1133384

Rev. 1

Comment

In the manuscript “Seaweed polysaccharide based products and materials: an assessment on their production from sustainability point of view” by Chudasama et al. the authors review the literature articles concerning the production and usage of different polysaccharides obtained from sea products.

The paper is well written, clear but some corrections are needed:

  1. It will be of interest if the authors can indicate the total amount of produced marine polysaccharides in the last years (tones/year)

Our Response

The authors acknowledge the reviewer for valuable suggestions that helps to improve the manuscript. The following data is added in the revised manuscript as per reviewer’s suggestions.

Kappaphycus alvarezii and Eucheuma denticulatum are the two seaweed species which contribute to 88% of raw material for carrageenan production. Malaysia, Indonesia and Phillipines produce around 1,20,000 tons per year. The world carrageenan production increased from less than 1 million wet tones in 2000 to more than 9 million wet tones in 2014 [9]. The global Carrageenan market was estimated to be around 931.6 million USD in 2020 and its valuation is expected to reach 1.2 billion USD at a CAGR (Compound Annual Growth Rate) of 5.6 % by 2025 [10]. China is the world leader in agar agar pro-duction with harvest of 2.7 million tonnes of farmed Gracilaria seaweed (main source of agar) in 2015 [11]. In 2015, the global agar market was estimated to be USD 214.98 mil-lions which is further anticipated to grow at 4.9% CAGR from 2016 to 2025 [12]. The al-ginate production is estimated to be around 30,000 metric tonnes annually which comes from farmed brown seaweed of the genera Laminaria and Macrocystis [13]. The global alginate market size was valued at USD 728.4 million in 2020 and is expected to grow at a compound annual growth rate (CAGR) of 5.0% from 2021 to 2028 [14].

  1. Hessami, M.J., Salleh, A. and Phang, S.M., Bioethanol a by-product of agar and carrageenan production industry from the tropical red seaweeds, Gracilaria manilaensis and Kappaphycus alvarezii. Iranian Journal of Fisheries Sciences. 2020, 19(2), 942-960, doi.org/10.22092/ijfs.2018.117104.
  2. Market Data Forecast. Available online: https://www.marketdataforecast.com/market-reports/carrageenan-market (accessed on 15 March 2021)
  3. Globefish Research Programme. Available online: http://www.fao.org/3/ca1121en/CA1121EN.pdf (accessed on 15 March 2021)
  4. Grand view research. Available online: https://www.grandviewresearch.com/industry-analysis/global-agar-agar-gum-market (accessed on 15 March 2021)
  5. Hay, I.D., Rehman, Z.U., Moradali, M.F., Wang, Y. and Rehm, B.H., Microbial alginate production, modification and its applications. Microbial biotechnology. 2013, 6(6), 637-650. doi.org/10.1111/1751-7915.12076
  6. Grand view research. Available online: https://www.grandviewresearch.com/industry-analysis/alginate-market (accessed on 15 March 2021)

Comment

  1. the number of recent articles is very scarce therefore some recent references must be added, such as: https://dx.doi.org/10.3390%2Fmolecules25143152; https://doi.org/10.3390/polym13030477; https://doi.org/10.1016/j.progpolymsci.2017.06.001; https://doi.org/10.1007/s12272-017-0958-2; https://doi.org/10.3389/fpls.2020.554585; https://dx.doi.org/10.3390%2Fmd18010007; http://dx.doi.org/10.3390/md18120658; but there are many more and the authors must indicate their originality in comparison with ones of these reviews.

Our Response

The authors thank the reviewer for the inputs and several recent articles are now updated in the revised manuscript as per suggestions.

“The polysaccharides with specific and distinguished characteristics are isolated from the vast marine source and such polysaccharides are absent in terrestrial plants. These algal polysaccharides serve as fascinating tools for therapeutic and industrial applica-tions which include nutraceuticals, pharmaceuticals, cosmeceuticals and functional foods. At present such bioactive sulphated polysaccharides are used as formulations in feed, food and pharmaceutical applications but their various biological capacities attract researchers [3,4]. Marine polysaccharides have also been explored as efficient drug de-livery systems combining the advantages of micelles and natural biopolymers showing excellent compatibility, biodegradability, non-toxicity, non-immunogenicity, extended blood circulation time and better drug loading capacity [5-7].

Polysaccharides from the marine sources have found their value addition due to their unprecedented therapeutic properties and hence are exploited in tissue engi-neering, stent coating and biomolecules immobilization [8].”

  1. Hentati, F., Tounsi, L., Djomdi, D., Pierre, G., Delattre, C., Ursu, A.V., Fendri, I., Abdelkafi, S. and Michaud, P., Bioactive Polysaccharides from Seaweeds. Molecules. 2020, 25(14), 3152, doi.org/10.3390/molecules25143152.
  2. Bilal, M. and Iqbal, H., Marine seaweed polysaccharides-based engineered cues for the modern biomedical sector. Marine drugs. 2020 18(1), 7, doi.org/10.3390/md18010007.
  3. Atanase, L.I., Micellar Drug Delivery Systems Based on Natural Biopolymers. Polymers. 2021 13(3), 477, doi.org/10.3390/polym13030477.
  4. Atanase, L.I., Desbrieres, J. and Riess, G., Micellization of synthetic and polysaccharides-based graft copolymers in aqueous media. Progress in Polymer Science. 2017, 73, 32-60, doi.org/10.1016/j.progpolymsci.2017.06.001.
  5. Zhong, H., Gao, X., Cheng, C., Liu, C., Wang, Q. and Han, X., The Structural Characteristics of Seaweed Polysaccharides and Their Application in Gel Drug Delivery Systems. Marine Drugs. 2020, 18(12), 658, doi.org/10.1016/j.progpolymsci.2017.06.001.
  6. Lee, Y.E., Kim, H., Seo, C., Park, T., Lee, K.B., Yoo, S.Y., Hong, S.C., Kim, J.T. and Lee, J., Marine polysaccharides: therapeutic efficacy and biomedical applications. Archives of pharmacal research. 2017, 40(9), 1006-1020, doi.org/10.1007/s12272-017-0958-2.

Reviewer 2 Report

I have reviewed the manuscript titled “Seaweed Polysaccharide Based Products and Materials: An Assessment on Their Production from a Sustainability Point of View’ and have the following comments  

The title suggest  that this review is based on new methods that are perhaps more sustainable than existing methods; or compare existing methods  for their sustainability. This brings us to the point on asking; What are the criteria used to measure sustainability?.

I found that the only place which addressed the sustainability issue was to do with Ionic liquids as shown below

Lines 277-280: Since a lower usage level of chemicals and better yield is expected from a commercial point of view, ILs and DESs can be considered as better solvents for sustainable production of seaweed polysaccharides. This statement requires more than lower usage and better yields to determine sustainability.  The authors need to inform us whether these ionic liquids meet the criteria versus existing processes.

Most figures: Do they represent a bench scale or industrial scale process? The latter is preferable as experiments are done at a miniscule scale and therefore to make an environmental impact one has to look at what occurs at an industrial scale.

In fact about half the manuscript covers the properties and use of seaweed-derived polysaccharides- this information is readily available in a number of recent reviews.

The ‘new material’ is about the derivatisation of the polysaccharides for new functionalities and the use of ionic liquids. Therefore I believe that they should review this specific area and be more critical in their analysis of this area. The sustainability angle can be left out unless that can show real measurable differences.

Other minor points:

Fig 1 ‘Extractive’ should be replaced by ‘Extract’

Typo: …mass washedmwith IPA

Line 79 & 81 seaweed are not weeds!

Author Response

Response to the Reviewer’s comments: molecules-1133384

Rev. 2

Comment

I have reviewed the manuscript titled “Seaweed Polysaccharide Based Products and Materials: An Assessment on Their Production from a Sustainability Point of View’ and have the following comments  

The title suggest that this review is based on new methods that are perhaps more sustainable than existing methods; or compare existing methods for their sustainability. This brings us to the point on asking; What are the criteria used to measure sustainability?

Our response

The major criteria considered to assess sustainability are ease of processing of seaweeds, minimum effluent production, and better yield of the products in comparison to conventional processes. Further, lesser usage of hazardous or non-eco-friendly chemicals or usage of bio-based chemicals which are more environment friendly in addition to the use of solvents which are recyclable or hydrated required to make the processes economically more feasible were another criteria considered for making the proposed processes sustainable. The proposed processes may ensure long-term profit-ability in the processing business and will help for better industrial level exploitation of seaweeds.

Although herein the results are based on the laboratory scale experiments and scale up using ionic liquids or deep eutectic solvents should be done in industrial level to make final techno commercial evaluation of the processes. Now this is written in the manuscript with a special focus to sustainability such that the readers understand the sustainability proposed in the article more easily. The below existing paragraphs are also rewritten focusing sustainability.  

“As described above, the extraction of polysaccharides from seaweed by conventional extraction process involve number of steps also produces effluents. The excessive amount of alkali and acid are used to break the cell walls of the seaweeds and the unutilized chemicals remained as effluent at the end of the process. Moreover, conventionally agar is extracted from agarophytes employing pre-treatment (alkaline) of the seaweed followed by autoclaving and subjecting the aqueous extract to several cycles of freeze-thaw to isolate the product followed by purifying the product further through solvent/chemical treatment and/or chromatography to eliminate residual impurities [113]. The agar thus obtained is purified by chromatography to obtain agarose. Considering the lengthy purification and expensive isolation process as well as requirement of expensive infrastructure, the process for the isolation of agar and preparation of agarose is not attractive for small-scale industries. Since ionic liquids are found to have very good dissolution, extraction and reuse efficiency, few of the imidazolium and ammonium based ILs were found to be suitable for the extraction of agar from Gracilaria dura, a red seaweed [114]. In one of the attempts, a bio-based ionic liquid namely choline laurate was found to have a selective affinity towards agarose present in the hot seaweed extracts resulting in eventual selective precipitation of the biopolymer as depicted photographically in Figure 9 [115]. This method may be considered as sustainable since agarose was directly obtained from the seaweed extract without involving any purification step of agar and freezing-thawing step which are energy intensive. The ionic liquid was re-cyclable and reusable for subsequent batches of experiments. 

Furthermore, few hydrated DESs were able to extract k-carrageenan from Kappaphycus alvarezii more efficiently in comparison to the conventional extraction methods [116]. The yield of the polysaccharide was about 60% in comparison to about 36% obtained by conventional process. From Figures 7 and 8 it is observed that, ILs [117] and hydrated DESs [116] were used to extract k-carrageenan from Kappaphycus alvarezii without using any alkaline treatment. The isolation of the polysaccharide was much easier in comparison to the conventional method and the quality of the polysaccharide was at par with their commercial counterparts.

  1. Wang, T.-P.; Chang, L.-L.; Chang, S.-N.; Wang, E.-C.; Hwang, L.-C.; Chen Y.-H.; Wang, Y.-M. Successful preparation and characterization of biotechnological grade agarose from indigenous Gelidium amansii of Taiwan. Process Biochem. 2012, 47, 550-554, doi.org/10.1016/j.procbio.2011.12.015.
  2. Effect of Sodium Sulfate on the Gelling Behavior of Agarose and Water Structure inside the Gel Networks. Tejwant Singh, Ramavatar Meena, and Arvind Kumar. J. Phys. Chem. B. 2009, 113 (8), 2519–2525, doi.org/10.1021/jp809294p.
  3. Sharma, M.; Chaudhary, J.P.; Mondal, D.; Meena, R.; Prasad, K. A green and sustainable approach to utilize bio-ionic liquids for the selective precipitation of high purity agarose from an agarophyte extract. Green Chem. 2015, 17, 2867-2873.
  4. Das, A.K.; Sharma, M.; Mondal D.; Prasad, K. Deep eutectic solvents as efficient solvent system for the extraction of k-carrageenan from Kappaphycus alvarezii. Carbohydr. Polym. 2016, 136, 930-935.
  5. Das, A.K.; Sequeira, R.A.; Maity, T.K.; Prasad K. Bio-ionic liquid promoted selective coagulation of k-carrageenan from Kappaphycus alvarezii extract. Food Hydrocoll.  2021, 111, 106382.

Comment

I found that the only place which addressed the sustainability issue was to do with Ionic liquids as shown below

Lines 277-280: Since a lower usage level of chemicals and better yield is expected from a commercial point of view, ILs and DESs can be considered as better solvents for sustainable production of seaweed polysaccharides. This statement requires more than lower usage and better yields to determine sustainability.  The authors need to inform us whether these ionic liquids meet the criteria versus existing processes.

Our Response

The sustainability issue with special reference to ionic liquid is now discussed in the revised manuscript.

Comment

Most figures: Do they represent a bench scale or industrial scale process? The latter is preferable as experiments are done at a miniscule scale and therefore to make an environmental impact one has to look at what occurs at an industrial scale.

Our Response

The figures shown in the manuscript are based on the laboratory scale experiments and the bench scale experiments using the green solvents are not conducted and it will be done in  the future.

In fact, about half the manuscript covers the properties and use of seaweed-derived polysaccharides- this information is readily available in a number of recent reviews.

The ‘new material’ is about the derivatisation of the polysaccharides for new functionalities and the use of ionic liquids. Therefore, I believe that they should review this specific area and be more critical in their analysis of this area. The sustainability angle can be left out unless that can show real measurable differences.

Our Response

We agree to the Reviewer’s view. But at this point of time our sole aim is to demonstrate a sustainable process for the extraction of seaweed polysaccharides and possible derivatization. The tables providing details of the sources, origin, structure as improved now by incorporating more information. 

Comment

Other minor points: 

Fig 1 ‘Extractive’ should be replaced by ‘Extract’

Typo: …mass washed with IPA

Line 79 & 81 seaweed are not weeds!

Our response

The authors acknowledge the reviewers to point out errors and these are corrected in the revised manuscript.

Reviewer 3 Report

Nishith A Chudasama et al. provide review of Seaweed Polysaccharide Based Products and Materials and Their Production from a Sustainability Point of  View. After close evaluation of the manuscript I would siggest revision according to the next points:

  1. In Keywords: as soon as "seaweed polysaccharides;" appears in the title it is not rational to use the same term in Key word. Please consider to include other words e.g agar, carrageenan, etc.
  2. In Table 1: I suggest to include references supporting the information provided in the Table.
  3. Review is interesting but following sections should be updated with recommended literature (see below).
  4. In Section 1.1: the phrase "Apart from the use of agar in the food and beverage industries, this phycocolloid is also used for bioengineering and biomedical applications as a gelling agent" should be supported with references: https://doi.org/10.1016/j.ijbiomac.2020.05.123; https://doi.org/10.1016/j.carbpol.2017.01.078; https://doi.org/10.1016/j.foostr.2016.10.003; https://doi.org/10.1007/s11094-014-1004-z;  http://www.aensiweb.net/AENSIWEB/aejsa/aejsa/2009/130-134.pdf; https://doi.org/10.1007/s13197-010-0162-6; https://doi.org/10.1016/j.jfoodeng.2008.05.036; https://doi.org/10.1016/j.talanta.2020.121892
  5. In Section 1.2: I suggest to update it with next papers: https://doi.org/10.1016/j.ijbiomac.2020.06.180; https://doi.org/10.1007/s10853-009-3770-7; https://doi.org/10.1208/s12249-020-1633-3; https://doi.org/10.1007/s10924-019-01553-5
  6. In Section 1.3: I suggest to update it with next papers: https://doi.org/10.1016/j.carbpol.2014.11.063; https://doi.org/10.3109/10408444.2013.861798; https://doi.org/10.3109/10408444.2013.861797; https://doi.org/10.1016/j.carbpol.2019.04.021; https://doi.org/10.1016/j.carbpol.2013.12.008; https://doi.org/10.1080/07328303.2017.1347668; https://doi.org/10.3390/molecules24234286; ; https://doi.org/10.3390/md18110583; https://doi.org/10.2174/1381612825666190425190754
  7. Section 2: I suggest to update with next articles: https://doi.org/10.1016/j.biomaterials.2012.01.007; https://www.academia.edu/download/33693650/G0433944.pdf; https://doi.org/10.1016/j.powtec.2020.10.027; https://doi.org/10.1016/j.ijbiomac.2019.08.030; https://doi.org/10.1002/smll.200801375;  https://doi.org/10.1016/j.carbpol.2017.05.001; https://doi.org/10.1002/cssc.201701860; https://doi.org/10.1016/j.ijbiomac.2016.11.095; https://doi.org/10.1016/j.carbpol.2021.117642
  8. All above mentioned polysaccharides have application in biomedicine.  I suggest to include a section regarding pharmacokinetic of polysacharides ( https://doi.org/10.3390/md18110557) and discuss importance of this issue as well.
  9. Section 3 I suggest to update with next papers: https://www.cellulosechemtechnol.ro/pdf/CCT1-2(2012)/p.45-52.pdf; https://doi.org/10.1016/j.seppur.2017.06.055; https://doi.org/10.1080/01496395.2017.1394881
  10. The conclusion require revision based on updated literature data.

Author Response

Response to the Reviewer’s comments: molecules-1133384

Rev. 3

Comment

Nishith A Chudasama et al. provide review of Seaweed Polysaccharide Based Products and Materials and Their Production from a Sustainability Point of  View. After close evaluation of the manuscript I would siggest revision according to the next points:

  1. In Keywords: as soon as "seaweed polysaccharides;" appears in the title it is not rational to use the same term in Key word. Please consider to include other words e.g agar, carrageenan, etc.

Our response

The reviewer’s comments are considered and the keywords are included accordingly.

  1. In Table 1: I suggest to include references supporting the information provided in the Table.

Our response

We thank the Reviewer for the comment and the desired information is now included in the table.

  1. Review is interesting but following sections should be updated with recommended literature (see below).

Our response

The authors acknowledge the reviewer for recommending appropriate literature that improves the manuscript. The literature is updated accordingly in the revised manuscript.

  1. In Section 1.1: the phrase "Apart from the use of agar in the food and beverage industries, this phycocolloid is also used for bioengineering and biomedical applications as a gelling agent" should be supported with references: https://doi.org/10.1016/j.ijbiomac.2020.05.123; https://doi.org/10.1016/j.carbpol.2017.01.078; https://doi.org/10.1016/j.foostr.2016.10.003; https://doi.org/10.1007/s11094-014-1004-z;  http://www.aensiweb.net/AENSIWEB/aejsa/aejsa/2009/130-134.pdf; https://doi.org/10.1007/s13197-010-0162-6; https://doi.org/10.1016/j.jfoodeng.2008.05.036; https://doi.org/10.1016/j.talanta.2020.121892

Our response

The above references are now included with suitable citations in the text. The revised paragraph reads as “Agar based films with enhanced shelf-life have been explored for their suitability to re-place plastic based packaging materials [24]. It is essential to understand the enzymatic mechanisms and agar biosynthesis to help in selecting seaweed materials with extended gelling properties utilising molecular markers [25]. The rheological and thermal properties of agar change upon alkali treatment and agaropectin enhances the gelling ability of agar [26]. Agar hydrogels were formed by mixing of agar-agar with different com-pounds such as glycerin, sorbitol, sodium citrate and sodium chloride with varying concentrations and their rheological behaviour was studied. Such studies are important to design hydrogels with required characteristics for specific applications [27]. Agar hydrocolloid is used as thickening and gelling agents for food applications and soft capsule preparation methods are also developed using agar as its base [28,29]. Agarose microparticles are explored to develop textural functionalities in beverages from liquid to fluid gels [30]. Agarose is used in the form of gel-based separation phase for micro-extraction due to easy fabrication, high inertness and biodegradability [31].”

  1. Mostafavi, F.S. and Zaeim, D., Agar-based edible films for food packaging applications-A review. International journal of biological macromolecules. 2020, 159, 1165-1176, doi.org/10.1016/j.ijbiomac.2020.05.123.
  2. Lee, W.K., Lim, Y.Y., Leow, A.T.C., Namasivayam, P., Abdullah, J.O. and Ho, C.L., Biosynthesis of agar in red seaweeds: A review. Carbohydrate polymers. 2017, 164, 23-30, doi.org/10.1016/j.carbpol.2017.01.078.
  3. Nishinari, K. and Fang, Y., Relation between structure and rheological/thermal properties of agar. A mini-review on the effect of alkali treatment and the role of agaropectin. Food structure. 2017, 13, 24-34, doi.org/10.1016/j.foostr.2016.10.003.
  4. Demchenko, D.V., Pozharitskaya, O.N., Shikov, A.N., Flisyuk, E.V., Rusak, A.V. and Makarov, V.G., Rheological study of agar hydrogels for soft capsule shells. Pharmaceutical Chemistry Journal, 2014, 47(10), 556-558, doi.org/10.1007/s11094-014-1004-z.
  5. Shikov, A.N., Pozharitskaya, O.N., Makarov, V.G. and Makarova, M.N., New technology for preparation of herbal extracts and soft halal capsules on its base. Am Eurasian J Sustain Agric, 2009, 3(2), 130-134.
  6. Saha, D. and Bhattacharya, S., Hydrocolloids as thickening and gelling agents in food: a critical review. Journal of food science and technology. 2010, 47(6), 587-597, doi.org/10.1007/s13197-010-0162-6.
  7. Ellis, A. and Jacquier, J.C., Manufacture and characterisation of agarose microparticles. Journal of food engineering. 2009, 90(2), 141-145, doi.org/10.1016/j.jfoodeng.2008.05.036.
  8. Tabani, H., Alexovič, M., Sabo, J. and Payán, M.R., An overview on the recent applications of agarose as a green biopolymer in micro-extraction-based sample preparation techniques. Talanta. 2020, 121892, doi.org/10.1016/j.talanta.2020.121892.

  1. In Section 1.2: I suggest to update it with next papers: https://doi.org/10.1016/j.ijbiomac.2020.06.180; https://doi.org/10.1007/s10853-009-3770-7; https://doi.org/10.1208/s12249-020-1633-3; https://doi.org/10.1007/s10924-019-01553-5

Our response

The above references are now included and the paragraph now reads as “They exhibit excellent properties such as water solubility, biodegradability, film forming ability and biocompatibility. These unique properties of the natural polysaccharide make it useful in fields of healthcare food industry, catalysis, for water treatment, packaging and immobilisation of cells [44,45]. This biodegradable polymer has also found its ap-plication in tissue engineering and preparation of biomaterial scaffolds that are very important for rendering medical needs [46]. Recently, alginic acid has been studied for its structural, molecular and functional abilities such as tablet ability, elasticity, deformation ability, disintegration ability and compressibility [47].”

  1. Guo, X., Wang, Y., Qin, Y., Shen, P. and Peng, Q., Structures, properties and application of alginic acid: A review. International Journal of Biological Macromolecules. 2020, 162, 618-628, doi.org/10.1016/j.ijbiomac.2020.06.180.
  2. Rani, P., Pal, P., Panday, J.P., Mishra, S. and Sen, G., Alginic acid derivatives: synthesis, characterization and application in wastewater treatment. Journal of Polymers and the Environment. 2019, 27(12), 2769-2783, doi.org/10.1007/s10924-019-01553-5.
  3. Sabir, M.I., Xu, X. and Li, L., A review on biodegradable polymeric materials for bone tissue engineering applications. Journal of materials science. 2009, 44(21), 5713-5724, doi.org/10.1007/s10853-009-3770-7.
  4. Benabbas, R., Sanchez-Ballester, N.M., Bataille, B., Leclercq, L., Sharkawi, T. and Soulairol, I., Structure-properties relationship in the evaluation of alginic acid functionality for tableting. AAPS PharmSciTech, 2020, 21(3),1-11, doi.org/10.1208/s12249-020-1633-3.

  1. In Section 1.3: I suggest to update it with next papers: https://doi.org/10.1016/j.carbpol.2014.11.063; https://doi.org/10.3109/10408444.2013.861798; https://doi.org/10.3109/10408444.2013.861797; https://doi.org/10.1016/j.carbpol.2019.04.021; https://doi.org/10.1016/j.carbpol.2013.12.008; https://doi.org/10.1080/07328303.2017.1347668; https://doi.org/10.3390/molecules24234286; ; https://doi.org/10.3390/md18110583; https://doi.org/10.2174/1381612825666190425190754

Our response

The above references are now included and the paragraph now reads as “In recent times, carrageenan-based biomaterials are gaining attention due to their multifunctional properties such as biodegradability, biocompatibility, non-toxicity, an-tiviral, antibacterial, anticoagulant, antioxidant, antitumor and immunomodulating properties [49,50]. Carrageenan has been widely exploited as food additives and several biomaterials for drug delivery applications have also been developed. These materials are also studied for their adverse effect on biological systems. The pH sensitivity and adhesive properties play an important role in preparation of such biomaterials [51-53]. These polysaccharides are source of sustainable and renewable polymers that can be employed for film formation and as coating materials. Blending these with other mate-rials to form composites enhance their film properties for potential applications. Such composites exhibit enhanced tensile strength and reduced hydrophilicity [54]. These sulphated polysaccharides have been exploited for their use in preparation of oral release tablets as a novel extrusion aid for the production of pellets and as a carrier stabilizer in micro/nanoparticles system. Due to its therapeutic properties, it is also used in tissue regeneration and cell delivery [55]. They are used as control release vehicles, gelling agents, encapsulating agents, beads, films and can efficiently encapsulate flavours, fragrances, enzymes and probiotics [56]. Recent study shows that these polysaccharides have anti-cancer activity thereby improving immunity and exhibit chemotherapeutic effects [57].”

  1. Qureshi, D., Nayak, S.K., Maji, S., Kim, D., Banerjee, I. and Pal, K., Carrageenan: a wonder polymer from marine algae for potential drug delivery applications. Current pharmaceutical design. 2019, 25(11), 1172-1186, doi.org/10.2174/1381612825666190425190754.
  2. Pacheco-Quito, E.M., Ruiz-Caro, R. and Veiga, M.D., Carrageenan: Drug Delivery Systems and Other Biomedical Applications. Marine Drugs. 2020, 18(11), 583, doi.org/10.3390/md18110583.
  3. Liu, J., Zhan, X., Wan, J., Wang, Y. and Wang, C., Review for carrageenan-based pharma-ceutical biomaterials: favourable physical features versus adverse biological effects. Carbohydrate Polymers. 2015, 121, 27-36, doi.org/10.1016/j.carbpol.2014.11.063.
  4. McKim, J.M., Food additive carrageenan: Part I: A critical review of carrageenan in vitro studies, potential pitfalls, and implications for human health and safety. Critical Reviews in Toxi-cology. 2014, 44(3), 211-243, doi.org/10.3109/10408444.2013.861797.
  5. Weiner, M.L., Food additive carrageenan: Part II: A critical review of carrageenan in vivo safety studies. Critical reviews in toxicology. 2014, 44(3), 244-269, doi.org/10.3109/10408444.2013.861798.
  6. Sedayu, B.B., Cran, M.J. and Bigger, S.W., A review of property enhancement techniques for carrageenan-based films and coatings. Carbohydrate polymers. 2019, 216, 287-302, doi.org/10.1016/j.carbpol.2019.04.021.
  7. Li, L., Ni, R., Shao, Y. and Mao, S., Carrageenan and its applications in drug delivery. Car-bohydrate polymers. 2014, 103,1-11, doi.org/10.1016/j.carbpol.2013.12.008.
  8. Chakraborty, S., Carrageenan for encapsulation and immobilization of flavor, fragrance, probiotics, and enzymes: A review. Journal of Carbohydrate Chemistry. 2017, 36(1), 1-19, doi.org/10.1080/07328303.2017.1347668.
  9. Liu, Z., Gao, T., Yang, Y., Meng, F., Zhan, F., Jiang, Q. and Sun, X., Anti-cancer activity of porphyran and carrageenan from red seaweeds. Molecules. 2019, 24(23), 4286, doi.org/10.3390/molecules24234286.

  10. Section 2: I suggest to update with next articles: https://doi.org/10.1016/j.biomaterials.2012.01.007; https://www.academia.edu/download/33693650/G0433944.pdf; https://doi.org/10.1016/j.powtec.2020.10.027; https://doi.org/10.1016/j.ijbiomac.2019.08.030; https://doi.org/10.1002/smll.200801375;  https://doi.org/10.1016/j.carbpol.2017.05.001; https://doi.org/10.1002/cssc.201701860; https://doi.org/10.1016/j.ijbiomac.2016.11.095; https://doi.org/10.1016/j.carbpol.2021.117642

Our response

The above references are now included and the paragraph now reads as “Alginates are important polysaccharides that form hydrogels at mild temperature and pH making them valuable industrial pharmaceutical excipient [70-72]. Biomaterials like nanospheres are prepared using alginates that exhibit drug delivery capacity and facilitate cell imaging. Alginate derivatives are also used for preparation of plastic ma-terials with tunable organosolubility and thermal properties [73-75]. Catalytic hydro-genation of the polysaccharide leads to sugar alcohols that may provide opportunities to diversify biomass resources [76].”

  1. Pawar, S.N. and Edgar, K.J., Alginate derivatization: a review of chemistry, properties and applications. Biomaterials. 2012, 33(11), 3279-3305, doi.org/10.1016/j.biomaterials.2012.01.007.
  2. Benabbas, R., Sanchez-Ballester, N.M., Bataille, B., Sharkawi, T. and Soulairol, I., Development and pharmaceutical performance of a novel co-processed excipient of alginic acid and microcrystalline cellulose. Powder Technology. 2021, 378, 576-584, doi.org/10.1016/j.powtec.2020.10.027.
  3. Soares, S.F., Rocha, M.J., Ferro, M., Amorim, C.O., Amaral, J.S., Trindade, T. and Daniel-da-Silva, A.L., Magnetic nanosorbents with siliceous hybrid shells of alginic acid and carrageenan for removal of ciprofloxacin. International journal of biological macromolecules. 2019, 139, 827-841, doi.org/10.1016/j.ijbiomac.2019.08.030.
  4. Guo, R., Li, R., Li, X., Zhang, L., Jiang, X. and Liu, B., Dual‐functional alginic acid hybrid nanospheres for cell imaging and drug delivery. Small. 2009, 5(6), 709-717, doi.org/10.1002/smll.200801375.
  5. Matsumoto, Y., Ishii, D. and Iwata, T., Synthesis and characterization of alginic acid ester derivatives. Carbohydrate polymer. 2017, 171, 229-235, doi.org/10.1016/j.carbpol.2017.05.001.
  6. Zia, K.M., Tabasum, S., Nasif, M., Sultan, N., Aslam, N., Noreen, A. and Zuber, M., A review on synthesis, properties and applications of natural polymer based carrageenan blends and composites. International journal of biological macromolecules, 2017, 96, 282-301, doi.org/10.1016/j.ijbiomac.2016.11.095.
  7. Ban, C., Jeon, W., Woo, H.C. and Kim, D.H., Catalytic Hydrogenation of Macroalgae‐Derived Alginic Acid into Sugar Alcohols. ChemSusChem. 2017, 10(24), 4891-4898, doi.org/10.1002/cssc.201701860.

  1. All above mentioned polysaccharides have application in biomedicine.  I suggest to include a section regarding pharmacokinetic of polysacharides ( https://doi.org/10.3390/md18110557) and discuss importance of this issue as well.

Our response

Considering substantial applications of polysaccharides in pharmaceuticals, we intend to write the pharmaceutical potential of the seaweed polysaccharides separately in another review article. We believe the inclusion of pharmaceutical aspects of the polysaccharides may confuse the readers of this review article which primarily talks about the production of the seaweed polysaccharides and their origin.

  1. Section 3 I suggest to update with next papers: https://www.cellulosechemtechnol.ro/pdf/CCT1-2(2012)/p.45-52.pdf; https://doi.org/10.1016/j.seppur.2017.06.055; https://doi.org/10.1080/01496395.2017.1394881

Our response

The above references are now included and the paragraph now reads as “Ionic liquids such as 1-butyl-3- methylimidazolium chloride with co-solvent dimethyl sulfoxide was used for the fabrication of agar/biopolymer blend aerogels [118]. Several methods have been developed for the extraction of seaweed polysaccharides, ionic liq-uids are also explored recently for their carrageenan extraction ability from red seaweed [118,119].”

  1. Shamsuri, A.A., Abdullah, D.K. and Daik, R., Fabrication of agar/biopolymer blend aerogels in ionic liquid and co-solvent mixture. Cell. Chem. Technol. 2012, 46(1-2), 45-52.
  2. Chew, K.W., Juan, J.C., Phang, S.M., Ling, T.C. and Show, P.L., An overview on the development of conventional and alternative extractive methods for the purification of agarose from seaweed. Separation Science and Technology. 2018, 53(3), 467-480, doi.org/10.1080/01496395.2017.1394881

  1. The conclusion require revision based on updated literature data.

Our response

The conclusion is revised as desired by the Reviewers.

Reviewer 4 Report

Reviewer comments and suggestions

The current review discussed seaweed polysaccharides, their origin, and extraction from different sources of seaweeds, application, and chemical modification. The author has discussed the process of derivatization of the polysaccharides that lead to new functionalities by chemical modification such as esterification, amidation, amination, C-N bond formation, sulphation, acetylation etc. Moreover, they reviewed ionic solvent systems from a sustainability point of view and prospects for efficient extraction and functionalization of seaweed polysaccharides. The manuscript required proofreading of language correction. Apart from this modification, the authors need to incorporate and defend these below comments at the time of submitting the revised version of the manuscript.  

The manuscript needs to be revised based on the below comments 

  1. Line 14, no need to use “also”
  2. Line 15-16 The sentences need to reform
  3. Line 17-19 Need to rewrite the sentence
  4. Line 24, please mention specific ( about these polysaccharides)
  5. The author has to use a citation followed by full stop. Check the format based on journal guideline
  6. Line 36-38 need references
  7. Please add more information to your table as these were very common and well known (line 42)
  8. need a reference (line 51)
  9. Line 67-68 You need to put references for this ray diagram or it is originally prepared by authors
  10. You need to explore more on various polysaccharides such as fucoidan, chitosan, ascophyllan etc.
  11. Line 120, chitin structure 
  12. the author needs to provide some examples with reference Line 127-128
  13. Line 129 Fucoidan, ascopllyan, prophyran etc
  14. “A number of polysaccharide derivatives are widely used in various fields”. Please do not write this kind of sentences

15.Line 142-144 Need references

16.I am not convinced with the figure provided figure 2, 3. Try to provide more information in the figures

17.Provide the usefulness of grafting

18.Line 177 There was no information on immunomodulatory activity

19.Line 228 , 218 redundant

20.Line 234-235 explore it more

21.Line 297 Section is short and that's why the author needs to discuss adequately.

22.Should be short and informative (future prospect)

Author Response

Response to the Reviewer’s comments: molecules-1133384

Rev. 4

The current review discussed seaweed polysaccharides, their origin, and extraction from different sources of seaweeds, application, and chemical modification. The author has discussed the process of derivatization of the polysaccharides that lead to new functionalities by chemical modification such as esterification, amidation, amination, C-N bond formation, sulphation, acetylation etc. Moreover, they reviewed ionic solvent systems from a sustainability point of view and prospects for efficient extraction and functionalization of seaweed polysaccharides. The manuscript required proofreading of language correction. Apart from this modification, the authors need to incorporate and defend these below comments at the time of submitting the revised version of the manuscript.  

 The manuscript needs to be revised based on the below comments 

  1. Line 14, no need to use “also”

Our response

The sentence is corrected according to reviewer’s suggestions in the revised manuscript.

  1. Line 15-16 The sentences need to reform

Our response

The sentences are reformed in the revised manuscript as suggested by the reviewer.

  1. Line 17-19 Need to rewrite the sentence

Our response

The sentence is corrected according to reviewer’s suggestions in the revised manuscript.

  1. Line 24, please mention specific (about these polysaccharides)

Our response

The polysaccharides are mentioned according to reviewer’s suggestions in the revised manuscript.

  1. The author has to use a citation followed by full stop. Check the format based on journal guideline

Our response

The format has now been corrected in the revised manuscript.

  1. Line 36-38 need references

Our response

Proper references have now been included in the revised manuscript.

  1. Please add more information to your table as these were very common and well known (line 42)

Our response

It is now provided

  1. need a reference (line 51)

Our response

Proper references have now been included in the revised manuscript.

  1. Line 67-68 You need to put references for this ray diagram or it is originally prepared by authors

Our response

The diagram depicted in the manuscript is originally prepared by the authors.

  1. You need to explore more on various polysaccharides such as fucoidan, chitosan, ascophyllan etc.

Our response

Our aim is to explore agar, carrageenan and alginates in the current review and hence the details of above polysaccharides are not included.

  1. Line 120, chitin structure 

Our response

The sentence is corrected according to reviewer’s suggestions in the revised manuscript.

  1. the author needs to provide some examples with reference Line 127-128

Our response

The sentence is corrected according to reviewer’s suggestions in the revised manuscript.

  1. Line 129 Fucoidan, ascopllyan, prophyran etc

Our response

The suggested examples are added in the revised manuscript.

  1. “A number of polysaccharide derivatives are widely used in various fields”. Please do not write this kind of sentences

Our response

This sentence is removed according to reviewer’s suggestions in the revised manuscript.

  1. Line 142-144 Need references

Our response

Proper references have now been included in the revised manuscript.

  1. I am not convinced with the figure provided figure 2, 3. Try to provide more information in the figures

Our response

It is now provided

  1. Provide the usefulness of grafting

Our response

It is now provided

  1. Line 177 There was no information on immunomodulatory activity

Our response

Suitable reference is now provided

  1. Line 228 , 218 redundant

Our response

Suitable reference is now provided

  1. Line 234-235 explore it more

Our response

Suitable reference is now provided

  1. Line 297 Section is short and that's why the author needs to discuss adequately.

This is now discussed with Suitable reference is now provided

  1. Should be short and informative (future prospect)

Our response

The conclusion is revised as desired by the Reviewers

Round 2

Reviewer 1 Report

The paper can be published as it is.

Author Response

Manuscript ID: molecules-1133384

The paper can be published as it is.

Our response

We thank the Reviewer for the constructive comments which has certainly improved the quality of the review article

Reviewer 2 Report

The authors have not adequately answered the question of sustainability in a comprehensive manner. As stated earlier he majority of the manuscript is on  the propertied and uses of marine -alga derived polysaccharides. There is one paragraph on the use of ILs and DESs and these come from about 3 publications. 

Furthermore the comparison of the sustainability features of the new process are not compared in any manner except to describe how these recyclable solvents replace higher energy  consuming systems or the use of alkali. However in 2 figures showing the 'sustainable processes' one (Fig 6) still has the use of alkali together with autoclaving while Fig 7 has the autoclaving step; whereas Fig 8 leaves out the early extractions steps completely.

Author Response

Manuscript ID: molecules-1133384

Comment

The authors have not adequately answered the question of sustainability in a comprehensive manner. As stated earlier he majority of the manuscript is on  the propertied and uses of marine -alga derived polysaccharides. There is one paragraph on the use of ILs and DESs and these come from about 3 publications. 

Furthermore the comparison of the sustainability features of the new process are not compared in any manner except to describe how these recyclable solvents replace higher energy  consuming systems or the use of alkali. However in 2 figures showing the 'sustainable processes' one (Fig 6) still has the use of alkali together with autoclaving while Fig 7 has the autoclaving step; whereas Fig 8 leaves out the early extractions steps completely.

Our response

There are 5 references mentioned for describing the extraction of seaweed polysaccharides using ionic solvents as a sustainable solvent medium and now more detailed discussion of the references are done. The comparison of the sustainability aspects is now done as desired by the Reviewer.

Furthermore, we want to mention here that, from our last 20 year research on extraction of seaweed polysaccharides from seaweed biomass we have observed below problems being faced by the seaweed processing industries which are major bottle necks for the sustainable production of the seaweed polysaccharides

  • Incomplete utilization of the biomass resulting production of residual solid waste
  • Use of excessive acid and alkali and generation of effluent with high BOD and COD
  • Requirement of huge amount of water for the processing

Hence a sustainable process for extraction of the polysaccharide is required which will involve less operational steps, use lesser amount of chemicals (alkali/acids, solvent), or use recyclable solvent systems or produce effluents which are biodegradable and safe for disposal. We tried to address this issues in the review article.

Please note, as shown in Figure 7 as mentioned by the Reviewer, alkali treatment and autoclaving is done. It may be kindly noted that our aim is to reduce the number of steps in the extraction  process of agarose and also to reduce the usage of excessive chemicals which produce effluents during the processing. It may be further noted that, agar or agarose is present in highly sulphated form (precursor) known as mu agar (m-agar) in the agarophytes. Alkali treatment is must to make a chemical rearrangement and replacement of sulphate groups and formation  of 3,6-anhydro galactose unit. This is inevitable step for agar production from chemistry point of view, Further, it should be understood that agar can be only extracted under pressure at elevated temperature and hence autoclaving is inevitable. The novelty we have discussed is that, as per scheme 1, to get agar, the crude seaweed extract gel need to be purified by freezing thawing which is done at -20 oC for 48 hours and 3 cycles are required. This is very energy intensive process further to get agarose, agar need to be further purified to remove the charges, which is done by ion exchange chromatography, precipitation using chemicals such as poly ethylene glycol. This is again tedious due to the involvement of several steps. On the other hand in Figure 7, we were able to isolate agarose (not agar) directly at room temperature without going through freezing/thawing, chromatographic purification  of agar. This method may be considered as sustainable since agarose was directly obtained from the seaweed extract without involving any purification stage of agar and freezing-thawing step which are energy intensive (Scheme 1). The ionic liquid was recyclable and reusable for subsequent batches of experiments. Isopropyl alcohol (IPA) used to wash the final product was also recovered by distillation and reused in the process. The process ensures lower energy inputs in the form of presence of lesser number of operations resulting lesser duration of the process. Further, recyclability and reusability of solvents ensures use of lesser amounts of chemicals and this make the process sustainable.  This is now discussed in the revised review.  

As mentioned above, extraction of seaweed polysaccharide felicitates at high temperature and under pressure and hence in Figure 8, autoclaving is done. But it can be seen from Scheme 2, to isolate carrageenan conventionally we need to treat the seaweed with alkali followed by autoclaving followed by pH adjustment by acid followed by filtration and finally precipitation using isopropyl alcohol. On the other hand the process we have demonstrated we were able to coagulate selectively carrageenan from the seaweed extract without alkali treatment or acid treatment. The IPA required in only half of the amount require for conventional process. This way the process is sustainable. As correctly observed by reviewer, we further tried to remove the autoclaving step which is energy intensive by using a better extraction solvent known as DES. As shown in Figure 9, no autoclaving required but carrageenan can be isolated in a much easier manner with higher in comparison to conventional process. This is now discussed in the revised review article.

Reviewer 3 Report

Authors have significantly revised paper and updated. However one issue still not addressed. Authors have mentioned in the manuscript the pharmacweutical application of polysaccharides: "...These algal polysaccharides serve as fascinating tools for therapeutic and industrial applications which include nutraceuticals, pharmaceuticals..."; "...At present such bioactive sulphated polysaccharides are used as formulations in 47 feed, food and pharmaceutical applications...", etc. Therefore I suggest to include a section regarding pharmacokinetic of polysacharides ( https://doi.org/10.3390/md18110557) and discuss importance of this issue as well.

Author Response

Manuscript ID: molecules-1133384

Comment

Authors have significantly revised paper and updated. However, one issue still not addressed. Authors have mentioned in the manuscript the pharmaceutical application of polysaccharides: "...These algal polysaccharides serve as fascinating tools for therapeutic and industrial applications which include nutraceuticals, pharmaceuticals..."; "...At present such bioactive sulphated polysaccharides are used as formulations in 47 feed, food and pharmaceutical applications...", etc. Therefore I suggest to include a section regarding pharmacokinetic of polysaccharides ( https://doi.org/10.3390/md18110557) and discuss importance of this issue as well.

Our response

We thank the reviewer for the suggestions. The desired information is included in the introduction section. However in order to maintain the flow, separate section was not included but the information including the above mentioned reference is adequately discussed in the introduction section. This now reads as

“As mentioned above, several seaweed polysaccharides are used in medicinal and pharmaceutical fields to impart various biological activities. Seaweed polysaccharide based nanoparticles, microspheres and gels were found to have sustained and controllable drug delivery potential for anti-cancer and anti-inflammatory drugs. [9] Several sulphated polysaccharides were found have cytotoxic properties and were used as antiviral substances against respiratory syncytial virus (RSV), Herpes simplex virus (HSV) types 1 and 2 and human immunodeficiency virus (HIV). Carrageenan has demonstrated as potential agent in vitro to impart antiviral activity like blocking of HIV and other sexually transmitted diseases [Carraguard (vaginal microbicide)]. [10] Biomedical and biological applications of seaweed polysaccharides i.e. alginate, carrageenan, fucoidan and ulvan displays in drug delivery, tissue engineering, biosensor and wound healing because of their gel forming properties and capacity of inducing important differentiation in stem cells. [11] Alginate is used for both immediate (for rapid absorption) and sustained (for reproducible and kinetically predictable) release of drug. Alginic acid and sodium salt of alginic acid are used as tablet disintegrant and tablet binding agent respectively for immediate drug release. Chitosan-alginate composite as well as calcium salt of alginic acid were used in wound healing and tolerance for diabetic foot lesion. In such applications, calcium from the alginate salt and sodium of the wound’s exudates go through ion exchange and form sodium alginate soluble gel, while free calcium ions helps in clotting and endowing the dressing. [12] Alginate and alginate beads were used in various treatments like in diabetes treatment by the transplantation of chondrocytes, hepatocytes, and islets of langerhans, treatment of gastroesophageal reflux disease and heartburn. [13-17] Combination of alginate, chitin/chitosan, and fucoidan gave a hydrogel sheet having favorable properties for wound healing in rats [18]. Similar to alginates, carrageenan is also used in drug delivery applications in various forms such as beads, gelling agent, nanoparticles, nano stabilizer, micro stabilizer, microspheres and in microcapsules [19, 20]. In the treatment of hypercholesterolemia, carrageenan can be used for sustained fluvastatin drug release and formulation [21]. The sustained release of ovalbumin macromolecule by the nanoparticles of chitosan-carrageenan is also reported [22].

As evident from above studies, seaweed polysaccharides are explored for their potential application as pharmaceutical drugs to treat number of ailments and hence it becomes very important to study their pharmacokinetics. This focuses study on the drug behaviour after its administration in the body systems that include absorption, distribution, metabolism and excretion (ADME). This helps to understand pharmacological activity at molecular level to determine correct doses, treatment methods and specific drug applications [23]. The parameters, which are important to gain knowledge on changes in drug concentrations with ADME, are apparent half-life of elimination (T1/2), the area under the curve (AUC), clearance (Cl), maximum concentration (Cmax) and time at which Cmax is observed (Tmax), median residual time (MRT), the high volume of distribution in the blood (Vss) and bioavailability (F). [24] Several analytical methods like biomarker assay, anti-Xa activity, gas chromatography, ELISA, HPLC, etc are used for the pharmacokinetic study of fucoidan. Furthermore, the pharmacokinetics of alginates and fucoidan seaweed polysaccharides are explored so far. [23] Despite the extensive exploitation of seaweed polysaccharides for their pharmaceutical applications, the pharmacokinetics of the same has not been explored much. Thus, an increase in this field of research is expected in the near future. ”

Reviewer 4 Report

No more comments

Author Response

Manuscript ID: molecules-1133384

No more comments

Our response

We thank the Reviewer for the constructive comments which has certainly improved the quality of the review article
